# Deletion of *ptn1*, a *PTEN*/*TEP1* Orthologue, in *Ustilago maydis* Reduces Pathogenicity and Teliospore Development

**DOI:** 10.3390/jof5010001

**Published:** 2018-12-20

**Authors:** Lalu M. K. Vijayakrishnapillai, John S. Desmarais, Michael N. Groeschen, Michael H. Perlin

**Affiliations:** Department of Biology, Program on Disease Evolution, University of Louisville, Louisville, KY 40292, USA; lalumkv2000@yahoo.com (L.M.K.V.); john.desmarais@louisville.edu (J.S.D.); michael.groeschen@louisville.edu (M.N.G.)

**Keywords:** *Ustilago maydis*, PTEN, teliospore development

## Abstract

The PTEN/PI3K/mTOR signal transduction pathway is involved in the regulation of biological processes such as metabolism, cell growth, cell proliferation, and apoptosis. This pathway has been extensively studied in mammals, leading to the conclusion that *PTEN* is a major tumor suppressor gene. *PTEN* orthologues have been characterized in a variety of organisms, both vertebrates and non-vertebrates, and studies of the associated PTEN/PI3K/mTOR pathway indicate that it is widely conserved. Studies in fungal systems indicated a role of *PTEN* in fungal defense mechanisms in *Candida albicans*, and in the developmental process of sporulation in *Saccharomyces cerevisiae*. The present study was aimed at investigating the role of the *PTEN* ortholog, *ptn1*, in *Ustilago maydis*, the pathogen of maize. *U. maydis ptn1* mutant strains where *ptn1* gene is deleted or overexpressed were examined for phenotypes associate with mating, virulence and spore formation. While the overexpression of *ptn1* had no substantial effects on virulence, *ptn1* deletion strains showed slight reductions in mating efficiency and significant reductions in virulence; tumor formation on stem and/or leaves were severely reduced. Moreover, tumors, when present, had significantly lower levels of mature teliospores, and the percent germination of such spores was similarly reduced. Thus, *ptn1* is required for these important aspects of virulence in this fungus.

## 1. Introduction

*Ustilago maydis*, is a plant pathogen responsible for smut disease in *Zea mays* [1]. In the past few decades, *U. maydis* has developed into a model system for a variety of research areas, including but not limited to, cellular transport processes, plant–pathogen interaction and DNA repair mechanisms [2]. Due to the high level of genetic similarity shared between *U. maydis* and metazoans, it may serve as a viable model for the study of essential cellular processes of higher eukaryotes [2,3]. Several homologues of human proteins that are involved in DNA repair and genomic stability are present in *U. maydis*, such as the BRCA2 orthologue, Brh2 [4].

*PTEN* (phosphatase deleted on chromosome 10; also known as *MMAC1* or *TEP1* (tensin-like phosphatase) is a human tumor suppressor gene mapping onto chromosome 10 at 10q23.3 [5]. The *PTEN* gene is either deleted or inactivated in several tumors, including those associated with breast, endometrial, brain, and prostate cancers [6]. The sequence of the phosphatase domain of *PTEN* is most similar to those of dual specificity protein phosphatases such as cell division cycle 14 (CDC14), baculovirus phosphatase (BVP), DUSP3, and DUSP4. However, the amino-terminal region of PTEN containing the phosphatase catalytic domain, also has homology to proteins such as actin-binding protein, tensin-1 (TNS1), and auxilin, the cofactor of ATPase heat shock protein HSC70, though such proteins are unrelated to PTEN catalytic functions [7]. The *PTEN* gene codes for a protein that is a major phosphatase enzyme that dephosphorylates PtdIns(3,4,5)P3 (PIP3) to PtdIns(4,5)P2 (PIP2), thereby counteracting the signaling by phosphatidylinositol-4,5-bisphosphate 3-kinase (PI3Ks) and playing a major role in the regulation of the PTEN/AKT/mTOR signaling pathway [8]. Studies conducted in mouse models and humans suggest that *PTEN* is highly regulated and very critical for cellular functions. It is evident from mouse model studies that a 20% reduction in *PTEN* expression level is enough to develop a range of tumors [9]. 

*PTEN* is mostly known for its tumor suppressor activity; however, it has also been shown to be involved in a variety of other functions, including the regulation of cell size, metabolism, ageing, and the development of neurons [10]. Orthologs of *PTEN* have been identified in *Drosophila*
*melanogaster* [11], *Caenorhabditis elegans* [12,13], *Candida albicans* [14], and budding yeast [15]. *PTEN* is essential for *Drosophila*, where it regulates cell size, cell proliferation, and apoptosis [11]. In *C. *elegans**, mutations in *PTEN* may cause disruptions in the developmental process, characterized by constitutive dauer (larval morphology) formation [12,13]. The deletion of the *PTEN* orthologue, *TEP1*, in the budding yeast, *S. cerevisiae*, revealed that it has a role in sporulation. *TEP1* mutants had defects in the trafficking or deposition of dityrosine, a major component of yeast spore walls [15]. In another yeast species, *C. albicans*, *PTEN* is involved in the defense mechanism, where it activates cofilin-1, an actin depolymerization factor during PGE2 (prostaglandin E2)-mediated inhibition of fungal phagocytosis by macrophages [14].

In this study, we have begun the functional characterization of the sole *PTEN* ortholog, *ptn1* (UMAG_03760)*,* in *U. maydis*. We examined the phenotypes of *U. maydis* cells overexpressing or deleted for *ptn1*. In the absence of a functional *ptn1* gene, *U. maydis* haploid cells grow normally in axenic culture, but displayed a reduced capability to tolerate cell wall stress. More strikingly, such mutants, when mixed with compatible mating partners, were defective in tumor formation, teliospore development, and the rate of teliospore germination.

## 2. Materials and Methods

### 2.1. Strains and Growth Conditions

The *U. maydis* strains used in this study are listed in Table 1. *Escherichia coli* strains TOP10 and DH5α (Invitrogen/Thermo Fisher) were used for all cloning and plasmid maintenance. *U. maydis* strains were grown at 26 °C in solid or liquid yeast extract-peptone-sucrose medium (YEPS), potato dextrose agar (PDA), Holliday salt solution (HSS) media [16], or Holliday Salt media, as described previously [17]. Strains grown in liquid media were shaken overnight at 250 rpm. Mating media was made with solid PDA and 1% activated charcoal, while stress media was made with solid YEPS and either Congo Red (Fisher Scientific, Fairlawn, NJ, 15 µg/mL), sodium chloride (1 M), calcofluor white (50 µM; Sigma-Aldrich, St. Louis, MO, USA) or sorbitol (1 M), as described in [18].

### 2.2. Vector Construction and Nucleic Acid Manipulations

The *ptn1* deletion strains in *U. maydis* were made, using homologous recombination by the DelsGate protocol, as described previously [22], using primers (Table 2) designated UPTN-K1-K4 to amplify the flanks of *ptn1*. The strains 1/2 Δ*ptn1* and 2/9 Δ*ptn1* were created by replacing the *ptn1* open reading frame (ORF) with the DelsGate deletion construct plasmid containing a carboxin resistance marker. Confirmed Δ*ptn1* mutants were complemented with a plasmid containing a hygromycin resistance gene, and a wild type copy of the gene under the control of the constitutive *otef* promoter (*P_otef_*), to yield 1/2 Δ*ptn1*c and 2/9 Δ*ptn1*c complemented strains. The overexpression strains of *ptn1* were generated by expressing the *ptn1* ORF under the control of P*_otef_*, using a derivative of p123 bearing a carboxin resistance marker [23]. P*_otef_*-based vectors were linearized using *Ssp*I digestion prior to transforming *U. maydis* protoplasts [23]. Additional *ptn1* deletion strains 1/2 Δ*ptn1*J, 2/9 Δ*ptn1*J, FB1 Δ*ptn1*J, SG200 Δ*ptn1*, and d132 Δ*ptn1* were generated via replacing the *ptn1* ORF with a hygromycin resistance cassette. The hygromycin replacement construct was prepared through overlap-PCR [24,25], using primers PTENUPLT, PTENUPRT, PTENDNLT, CPDNRT, HYGPTENLT, HYGPTENRT (Table 2) as described previously [24], with minor modifications. 

Transformed *U. maydis* strains were selected on YEPS agar plates containing either 3 μg carboxin per mL (Santa Cruz Biotechnology, Dallas, TX, USA) or 150 μg/mL hygromycin B (Gold Biotech). Transformation was confirmed using PCR with primers that would be diagnostic for true transformants, and distinct from bands for wild type (e.g., PTEN5/PTEN3). Also, for Δ*ptn1* derivatives, qPCR was used to confirm the loss of gene expression relative to the wild type. Primers were designed using the Primer3 software (http://bioinfo.ut.ee/primer3/) based on the genome sequence available at JGI (http://genome.jgi.doe.gov/Ustma1/Ustma1.home.html). PCR was carried out in a T100 Thermal Cycler (Bio-Rad Laboratories, Hercules, CA, USA). For PCR amplification reactions, either Hot Start ExTaq DNA Polymerase, PrimeSTAR HS DNA Polymerase (Takara Bio, USA), Platinum Taq Green Hot Start DNA Polymerase (Life Technologies Corporation), or EconoTaq DNA Polymerase (Lucigen Corporation) were used.

### 2.3. RNA Isolation and Expression Analysis

RNA isolation from *U. maydis* strains and quantitative real-time PCR were carried out, as described in [26]. The RNA expression data were analyzed in GraphPad Prism by using unpaired *T*-tests after normalization to wild-type expression under high ammonium conditions; a probability value of *p* < 0.05 was considered as a significant difference.

### 2.4. Cell Growth, Mating, and Plant Infection

For cell growth rate and mating assays, liquid cultures of the strains grown overnight in 4 mL YEPS liquid media shaken at 260 rpm were used at an absorbance at 600 nm (OD_600_) of between 0.6–0.8. To determine growth rates, cells grown overnight to reach exponential growth phase were collected, washed, and resuspended in fresh YEPS liquid media. Subsequently, 4 × 10^6^ cells were transferred into 10 mL of fresh YEPS liquid media, and optical density was determined every 2 h for the first 12 h and every 6 hr afterwards, up to 30 h [23,27]. Data were analyzed using Graphpad Prism. Mating assays were performed using *U. maydis* cells grown overnight in YEPS broth to an OD_600_ of 0.8, and cells were resuspended in water to an OD_600_ of 1.6 prior to spotting. A volume of 10 µL of opposite mating-type strains was co-spotted onto PDA charcoal plates, and the mating reaction was observed after incubation at 26 °C for 24–48 h [19]. 

For plant infection assay, 4 mL overnight cultures were diluted using 50 mL fresh YEPS liquid media and grown overnight to an OD_600_ of 1.0, harvested and resuspended in water to an OD_600_ of 3 prior to infection [20]. Plant infection was performed by inoculating 7–9 days old maize seedlings (Golden Bantam seeds, Bunton Seed Co., Louisville, KY and W. Atlee Burpee & Co., 62 Warminster, PA). Haploid strains of opposite mating types were mixed prior to plant infection, while diploid d132 and the solopathogenic SG200 strain were used directly, as previously described [19,28]. The virulence of each treatment group was scored by a disease index (DI) on a scale of 0 to 5, where 0 = No Symptoms, 1 = Chlorosis, 2 = Small leaf tumors, 3 = Large leaf tumors or small stem tumors, 4 = Large stem tumors with bending, and 5 = Death [19,29]. Thirty plants were used in each experiment, and the trials were repeated three times, with plants scored for disease symptoms on 14 and 21 days post-infection (dpi). A haploid *a1b1* background mutant strain crossed with a haploid *a2b2* mutant strain was compared in each case to the corresponding wild type cross. Additional measures of virulence included estimations of biomass, numbers of leaves, plant length, and chlorophyll content of infected plants. The dry biomass index of the infected plants was measured as previously described [30]. The above-ground parts of the infected plants where both partners were wild type, or both were mutant (i.e., Δ*ptn1* × Δ*ptn1* or *ptn1^otef^* × *ptn1^otef^*) were collected eight days post-infection (dpi), and washed once with tap water and twice with distilled water. Plants were blotted dry, and the samples were stored in labeled paper bags, and dried at 65 ± 5 °C in an incubator until a constant weight and dry biomass was recorded. Chlorophyll extraction was performed by the DMSO method, and contents were determined spectrophotometrically, as described previously [31,32] Plant infections were also performed on a dwarf variety of maize (Tom Thumb popcorn, High Mowing Organic Seeds, Walcott, VT) mainly to facilitate maize cob infections. For maize cob infection, maize plants were de-tasseled to increase the prospect of cob infection by preventing the pollination of ovules. Cobs were injected, as described previously [29,33]. In separate experiments, dwarf maize plants were infected with wild type (WT) and mutant strains on the leaf, following the protocol described [22].

### 2.5. Statistical Analysis

Statistical analysis of the disease index data from plant infection was conducted using Kruskal–Wallis Test with a Multiple Comparison Test using R software [34,35]. The data of the plant morphological characteristics such plant length, leaf number, dry biomass, chlorophyll content, spore germination rate. and spore count were analyzed using analysis of variance (ANOVA) followed by Tukey’s test [36] using GraphPad Prism software, and the probability values of *p* < 0.05 were considered as being significant.

## 3. Results

### 3.1. In Silico Analysis of ptn1

Initial analysis of the *U. maydis* genome sequence revealed the presence of a hypothetical-like gene (UMAG_03760), hereafter known as *ptn1*, predicted to encode a protein of 848 amino acids. The nucleotide sequence analysis (BLASTn) of the *ptn1* gene found the greatest similarity to the *PTENB* gene (XM_012330711.1) of *Pseudozyma hubeiensis*, a fungus also belonging to the order Ustilaginales. The *U. maydis* Ptn1 protein is characterized by the conserved protein tyrosine phosphatase catalytic (PTPc) domain motif (more specifically, the CDC14 subgroup of PTP proteins) present in other PTEN orthologues, as well as a phosphatase tensin-type domain, which is characteristic of all PTEN proteins (Figure 1, Appendix A). In silico prediction tools such as SignalP (http://www.cbs.dtu.dk/services/SignalP) and TargetP (http://www.cbs.dtu.dk/services/TargetP) suggested that Ptn1 is a cytoplasmic protein (Appendix A).

### 3.2. Deletion of ptn1 Led to the Reduced Production of the Aerial Hyphae Characteristics of the Mating Reaction on Charcoal Media 

Wild type, Δ*ptn1*, and overexpression strains (*ptn1*^otef^) were spotted onto PDA charcoal plates to assess any defects in mating or post-mating filamentation (Figure 2). Compatible wild type haploids (1/2 *a1b1* x 2/9 *a2b2*) produced dikaryotic hyphae that appeared as white ‘fuzzy’ growth when co-spotted. However, compatible *ptn1* deletion strains when co-spotted, showed a reduction in ‘fuzz’, especially when examined after the first 24 h. However, overexpression strains did not show a major change in ability to produce aerial hyphae. The complemented mutants, Δ*ptn1*c, were able to partially recover the wild type phenotype, indicative of successful mating.

To determine whether Δ*ptn1* cells produce pheromone, mating was performed using *U. maydis* tester strains, FBD11-7 and FBD12-17 (Figure 2). Such diploid strains are homozygous at the *a* mating locus, and they do not produce aerial hyphae when spotted alone on charcoal, but will become filamentous when they are plated together with cells that produce a different pheromone. The ‘fuzzy’ appearance appeared in the case of both the wild type *a1b1* and *a2b2* partners, together with their respective tester strains; on the other hand, there was a slight reduction in the production of aerial hyphae for the *ptn1* deletion strain in the *a2b2* background when paired with tester strain FBD11-7, suggesting reduced production of the a2 pheromone by this mutant. 

As an additional level of confirmation for the observed defects in the ‘fuzzy’ phenotype, the mating assay was performed using deletion strains in an additional genetic background (FB1). When mating of the wild type (1/2 × 2/9) was compared with that of *ptn1* deletion strains (1/2 Δ*ptn1* × 2/9 Δ*ptn1*, 1/2 Δ*ptn1*J × 2/9 Δ*ptn1*J*,* FB1 Δ*ptn1*J × 2/9 Δ*ptn1*J), similar results were obtained as in Figure 1, i.e., there was reduction in mating at 24 h for deletion strains, irrespective of the construct or the background (Figure 3A).

Formation of aerial hyphae on charcoal was also examined for *ptn1* deletion mutants in the diploid background (d132 Δ*ptn1*) and in the solopathogenic SG200 background (SG200 Δ*ptn1*). The deletion strain in the SG200 background did not show any change in the ‘fuzz’ intensity (Appendix A). However, the d132Δ*ptn1* strain showed reduced formation of aerial hyphae (Figure 3B), compared to the wild type. This result indicates that *ptn1* gene does show some level of haplo-insufficiency in the d132 diploid background, as has been observed for other genes in this species [28]. In contrast, the loss of *ptn1* appears at least partially compensated by the presence of the wild type allele in dikaryons as demonstrated by 2/9 x 1/2 Δ*ptn1*. 

### 3.3. *prf1* Is Upregulated in Δ*ptn1* under Low Ammonium Conditions

The lower than expected formation of aerial hyphae during the mating of the 2/9 Δ*ptn1 a2b2* with the tester strain, FBD 11-7 strain led us to speculate whether there was an alteration in the expression of mating genes affecting pheromone production. The fusion of haploid cells during mating leads to the differentiation of *U. maydis* into subsequent filamentous dikaryon [20]. Also, environmental conditions such as acidic pH, lipids, and low ammonium conditions can trigger the filamentous response in *U. maydis* [37,38,39], while acetate inhibits filamentation, due to mating or to exposure to oleic acid [40]. Hence, a qRT-PCR analysis was conducted for several genes involved in the mating pathway, e.g., *mfa2*, encoding a2 pheromone precursor, *pra2*, encoding pheromone receptor, and *prf1*, encoding a transcription factor [41]. The analysis involved comparing the expression levels between wild type 2/9 and 2/9 Δ*ptn1* under high and low ammonium conditions. The expression of all the strains were compared to the wild type 2/9 strain with high ammonium (Appendix A). The data indicated that the *prf1* levels decreased between high and low ammonium conditions in wild type, whereas it significantly increased in the *Δptn1* mutant from high to low ammonium. There was a significant difference between the relative expression of *prf1* between wild type under low ammonium, and *Δptn1* mutants under low ammonium. Even though there was no significant difference in the expression of *mfa2* and *pra2* under similar conditions, the trend showed that while the *mfa2* and *pra2* genes tended to be less well expressed under low ammonium conditions in wild type, for the *Δptn1* mutant, their expression appeared to be in the opposite direction.

### 3.4. Cell Growth of ptn1 Mutants Is Largely Unaffected, while Stress Tolerance Is Reduced under Certain Conditions

In standard growth media such as yeast extract-peptone-sucrose (YEPS) or potato dextrose broth (PDB), *ptn1* mutants were able to grow and showed no significant differences from wild type cells, in terms of growth rate (Appendix A). The effects of stressors, including high osmotic medium, i.e., YEPS supplemented with 1 M sorbitol or 1 M NaCl was tested, but not found to affect growth (data not shown). However, *ptn1* mutants showed a difference from wild type strains on YEPS media supplemented with Congo Red (CR) or calcofluor white (CFW). Both *ptn1* deletion and overexpression strains showed reduced tolerance when compared to the respective wild type strains (Figure 4A), while the complemented strains were able to partially recover the wild type growth (Figure 4B) in CR media, as well as on CFW media (Figure 4C). Overall, these experiments suggested that the mutation of *ptn1* has an impact on the integrity of the cell wall.

### 3.5. Pathogenicity Is Reduced by *ptn1* Deletion

Compatible opposite mating type strains were mixed and injected into maize seedlings to assess the effect of deletion of *ptn1* on *U. maydis* virulence. The results at 14 dpi are presented in the graph as a percentage of symptom formation in infected plants. Statistical analysis was performed on the disease index score (mean ± standard deviation) with a Kruskal–Wallis test and a multiple-comparison test. The results (Figure 5) indicate that deletion of *ptn1* results in reduced virulence compared to the wild type, and this reduction was statistically significant compared to the other infections. There were significantly more healthy plants surviving at the end of the study among the plants infected with *ptn1* deletion mutants. On the other hand, overexpression of *ptn1* did not yield significant changes in the virulence. For infections where both partners carried the *ptn1* deletion, the ability of the fungus to form tumors was severely reduced. While in infections with both wild type partners approximately 82% of infected plants had tumors, only roughly 30% of infections had tumors when both mating partners carried the Δ*ptn1* lesion. The complemented Δ*ptn1*c mating partners were partially able to rescue this defect. More plants (~90%) showed signs of infection in the Δ*ptn1*c infected group when compared to the Δ*ptn1*-infected plants (~68%). 

In a separate experiment, deletion strain d132 Δ*ptn1* also showed a reduced severity of infection compared to its wild type progenitor, indicating that one functional copy of *ptn1* was insufficient for full pathogenicity (Figure 5B). The phenotypic trend of reduced pathogenicity was confirmed by using deletion mutant in a SG200 background, though the difference (*p* = 0.05048) was only marginally significant (Appendix A). In addition to the above-mentioned infections using d132 background strains, the *ptn1* deletion strain in the *a1b1* background mixed with the WT strain in *a2b2* background was examined for the effect of virulence when one wild-type copy of the gene was present; i.e., the 1/2 *a1b1* Δ*ptn1* strain was crossed with the 2/9 *a2b2* strain, and vice versa. This was compared with wild type (1/2 × 2/9) infection and the deletion mutant infection (1/2 Δ*ptn1* × 2/9 Δ*ptn1*). The results are shown in Figure 6, and they indicate that the presence of one wild-type copy can rescue the phenotype to some extent. 

### 3.6. Pathogenicity Calculated by Additional Plant Growth Indicators

The effect of the deletion of *ptn1* on the virulence of the fungus was measured by a range of plant growth indicators. This provided a more comprehensive view of the extent of the effect on pathogenicity. In this experiment, plants infected with wild-type cross (1/2 × 2/9) strains were compared with those infected with deletion cross and overexpression cross strains. A total of 28 plants were used per treatment group, and the experiment was repeated a total of three times. The total number of leaves and the length of the plants were measured on the day of infection, as well as on 4 dpi (days post-infection) and 8 dpi. Measurements were not taken after 8 dpi, as many plants were drooping and unable to stand erect due to infection, thereby posing difficulties in making accurate length measurements. Both the Δ*ptn1* and *ptn1* overexpression infections yielded significantly taller plants (i.e., reduced infection) compared to the wild type (Figure 7A). The average number of leaves produced were significantly lower in plants infected with wild-type partners, compared to plants infected with overexpression strains or deletion strains, with the latter showing the highest average number of leaves (Figure 7B). The dry biomass index of the plants infected with wild type strains was significantly lower when compared to Δ*ptn1* infections. The *ptn1^otef^*-infected plants had a lower biomass index than the Δ*ptn1*-infected plants, but this was higher than for wild-type infections. However, the differences in biomass, plant height, or leaf length of the plants infected with the overexpression strain were not statistically significant when compared to wild type infection, which indirectly indicates that only the deletion of *ptn1* results in a significant alteration in the virulence of the fungus. 

During infection there was normally a color change in the leaves of infected plants, due to chlorosis. The wild-type infected plant leaves had more yellow coloration/chlorosis, compared to Δ*ptn1*-infected plants. Several papers have shown, for wild-type *U. maydis* infections, an increase in chlorosis and a decrease in the chlorophyll content of infected leaves [42,43]. Thus, chlorophyll estimation was performed to examine whether there was a corresponding change in the level of photosynthetic pigments such as chlorophyll *a* and chlorophyll *b* between the different treatment groups. The concentrations of both chlorophyll *a* and chlorophyll *b* pigments were significantly reduced in wild-type infected plants, when compared to comparable infections, using the deletion strain partners (Figure 8). The chlorophyll *a* and chlorophyll *b* data were used to calculate chlorophyll *a*/*b* ratio which is indicative of the level of stress in plants [44,45]. The chlorophyll *a*/*b* ratio for plants from the Δ*ptn1* infection was significantly higher compared to that for wild type. 

### 3.7. Impact of ptn1 Mutations on Spore Formation and Germination

The plants infected with *ptn1* deletion strains showed visibly reduced production in tumors in terms of size and number (Appendix A). Tumors were collected from all three treatment groups, 1/2 × 2/9, Δ*ptn1* × Δ*ptn1* and *ptn1*^otef^ × *ptn1*^otef^, and weighed. The experiment was repeated three times. The data indicated that the WT (1/2 × 2/9) fungus produced far more tumors in plants when compared to the deletion strain or the overexpression strain. In the case of deletion strain, tumors produced were significantly fewer than for the WT or overexpression strain infections (*p* = 0.011, Tukey’s test). Microscopic analyses of tumors from plants infected with compatible wild type strains or compatible Δ*ptn1* strains, showed that the number of mature spores were relatively low in deletion strains (Appendix A). The teliospores from wild-type infections overall were spherical and darkly pigmented, and the number of mature teliospores increased in quantity with the age/size of tumor, becoming clearly visible without magnification. However, while the Δ*ptn1* × Δ*ptn1* infected plants produced some tumors, the quantity of darkly pigmented teliospores was less (Figure 9), and in most cases, they were not obviously visible. It was difficult to quantify the presence of immature spores, due to the inconsistency in the stages of infection between tumors, and even within different parts of the same tumor. However, observations made from multiple independent infections indicated that the number of spores matched the intensity of infection. Crosses between Δ*ptn1* derivatives constitutively expressing an ectopic copy of *ptn1* saw an improvement in tumor number and size, as well as spore number, but did not show a complete return to the wild type levels for these traits. Rather, the spore count from Δ*ptn1*c was closer to that of *ptn1*^otef^ than Δ*ptn1,* but it was still less compared to wild type (Figure 9).

The teliospores were extracted from tumors and allowed to germinate using previously published methods [46]. Tumors from maize plants inoculated with wild type cross (1/2 × 2/9), deletion strain cross (Δ*ptn1* × Δ*ptn1)*, and the cross of overexpression strains (*ptn1*^otef^ × *ptn1*^otef^) were used for germination studies. Since spore production was reduced in infections with the deletion strain cross, we examined whether there were also defects in spores from such infections. The Δ*ptn1* strains showed a significantly lower rate of germination when compared to that for wild type infections (*p <* 0.05, Tukey’s test); the complemented Δ*ptn1* derivatives could only partially overcome the defect in spore germination rates (Figure 10). 

The formation of germination tubes from the spores (Figure 11) was examined to see if such defects could explain the reduced germination rates observed. Spores collected from tumors of plants infected with 1/2 *a1b1* × 2/9 *a2b2* and Δ*ptn1 a1b1* × Δ*ptn1 a2b2* strains were plated onto 2% water agar plates. The formation of germination tubes was observed microscopically after 24 to 48 h.

In a pilot experiment to examine additional possible defects of the *ptn1* deletion crosses, plant infections were performed on maize ears of a dwarf variety of maize (*n* = 2) (Appendix A). In this way, we investigated whether the deletion of *ptn1* affected teliospore production when inoculated into the developing embryos (ears) of plants with mixtures of compatible wild type or mutant strains. Only the wild-type inoculation produced tumors, and these were filled with black teliospores (Appendix A); in contrast, the deletion strains did not show ear infection (Appendix A). As a control for differences in the infection rates on different varieties of maize, the same dwarf variety was infected on the stems of plants; deletion strains failed to yield infection progression beyond chlorosis, and over time, infected plants appeared to be as healthy as uninfected plants. However, the wild-type infection produced severe infection symptoms, with the death of most infected plants (not shown). 

## 4. Discussion

In this study we examined the role of the *PTEN* orthologue of *U. maydis, ptn1,* in the pathogenic program of the fungus and in teliospore formation. The predicted Ptn1 protein of *U. maydis* has a conserved protein tyrosine phosphatase catalytic motif (PTPc) in its N-terminal region, a motif present in the PTEN proteins of most other organisms (Appendix A). This is consistent with the finding that the PTEN orthologue of *Arabidopsis thaliana,* AtPTEN1, shares similarity with higher eukaryotes only in the N-terminal region where the catalytic domain is present [47]. However, the lack of conservation outside of the catalytic PTPc motif suggests that *ptn1* may respond mostly to signals that are unique to *U. maydis*. In our study, it was observed that a mutation in *ptn1* may lead to a decrease in tolerance to stress conditions, especially in media supplemented with cell wall stress agents such as Congo Red and calcofluor white. It has been observed in *S. cerevisiae* that dual-specificity protein tyrosine phosphatases such as Ptp2 and Ptp3 play a regulatory role in cell integrity pathways [48]. It has also been reported in mouse model studies that *PTEN* is involved in stress response pathways that are connected to oxidative stress [49]. 

In *U. maydis*, successful mating is indispensable for the formation of the dikaryon, which is the infectious form of the fungus [50]. Mating assays on charcoal plates and subsequent aerial hyphae indicative of infectious filament formation were used to measure this prerequisite for successful infection. The *ptn1* deletion strains had reduced fuzz formation during the initial 24 hr. Since the difference was unnoticeable after 48 h, this indicates that the deletion caused delay in the onset of mating, and thus, only a minor defect in the mating capability. The RNA expression pattern of *mfa2* and *pra2* under low ammonium conditions could not explain the reduced aerial hyphal production in matings involving 2/9 Δ*ptn1.* On the other hand, we did observe the altered expression of *prf1* in the Δ*ptn1 a2b2* mutant. Since Prf1 is a transcription factor, its dysregulation could alter the expression of some unidentified downstream targets, leading to translational or post-translation effects on mating-specific targets. It has been reported before that overexpression of *ras1*(UMAG_11476) resulted in elevated levels of *mfa* (a pheromone precursor) in *U. maydis* [51]. A bioinformatic analysis looking for potential binding partners of Ptn1 has indicated that UMAG_11476 is a potential binding partner (Appendix A). Thus, it is plausible that UMAG_11476 is a potential downstream target for Ptn1. In humans, the dosage of *PTEN* is important for tumor suppression, as is evident from the data that monoallelic mutations (i.e., heterozygous) at the *PTEN* locus have been found in various cancer types, including prostate cancer, breast cancer, colon cancer, and lung cancer [52]. Our data from the mating and plant infection experiments using *ptn1* deletion strains in the d132 diploid background are in accord with this observation, i.e., the loss of one copy of the *ptn1* gene is only partially compensated by the presence of the wild-type allele in dikaryons and diploids. It has been observed that deletion strains of *biz1* gene in the SG200 background could filament normally, while showing reduced virulence in plant infection [53]. We observed a similar phenotype with SG200 Δ*ptn1*.

Infection of maize plants with compatible wild type strains (1/2 *a1b1* × 2/9 *a2b2*) and mutant strains (Δ*ptn1 a1b1* × Δ*ptn1 a2b2* or *ptn1*^otef^ × *ptn1*^otef^) showed that the *ptn1* deletion mutants were less virulent than wild type strains. Though the Δ*ptn1* mutants were able to infect maize plants, they showed reduced symptoms, especially in tumor number and size (Appendix A); *ptn1* overexpression did not significantly alter virulence. We were able to rescue the virulence partially by complementing the deletion strains with a wild type copy of the *ptn1* gene. Although deletion strains constitutively expressing *ptn1* ectopically were only partially rescued, the fact that independently-isolated *ptn1* deletion mutants, produced by different means and in different genetic backgrounds, had similar levels of defects in measures of pathogenicity (Appendix A), strongly supports the role of the *ptn1* gene in virulence for *U. maydis*. A study of the pathogenic fungus, *Thielaviopsis basicola*, causing black root rot disease of cotton plants, has shown that plant growth factors such as plant height, dry biomass, etc., are valid indicators of pathogen infection [54]. Similarly, biochemical indicators like chlorophyll content or chlorophyll (*a*/*b*) ratio can be used as indicators of level of fungal infection in plants, as shown with *Crinipellis perniciosa*, causing witches broom disease in cocoa plants [55] and *Colletotrichum lindemuthianum* infecting *Phaseolus vulgaris* (common bean) plants [56]. In oxeye daisy (*Leucanthemum vulgare*) plants infected with the fungus *Rhizoctonia solani*, an 81% reduction in shoot and a 70% reduction in root biomass were observed [57]. In our study, the reduction in pathogenicity was confirmed in measures of plant height, numbers of leaves, and levels of chlorophyll content. The plants infected with deletion strains were significantly healthier compared to the wild-type infected plants. 

The dissection and microscopic analysis of the tumors from plants infected with Δ*ptn1* mutants showed that while tumors were produced, a large proportion of those tumors lacked black teliospores. Moreover, tumors from infections with the deletion strains had significantly fewer spores than tumors from wild-type infections. The spore germination assay found a significantly lower rate of germination of spores from infections by the Δ*ptn1* mutants. However, there was no obvious difference between the wild-type and mutant strains in germination tube formation.

Effects of the *ptn1* deletion in vitro, especially in haploid or diploid sporidia, were subtle (Figure 2 and Figure 3). A slight reduction in aerial hyphae production on charcoal, and an increased sensitivity to cell wall stressors were seen for such mutants. A similar cell wall defect was reported for *ptn1 (PTEN* orthologue) mutants in fission yeast, *Schizosaccharomyces pombe* [58]. Our study indicates that while appressoria formation and penetration (not shown) were not impaired for the *ptn1* mutants, pathogenic development was obstructed at stages after penetration.

It has been noted in the case of some signaling proteins that aberrant expression in either direction can end up in same phenotypes. For example, when the tumor suppressor SRPK1 (serine/arginine-rich splicing factor kinase), which mediates the recruitment of the phosphatase PHLPP1 to Akt (besides its role in regulated splicing) was deleted or overexpressed, it resulted in the promotion of cancer [59]. This is especially a possibility when the same protein has multiple roles in interconnected signal transduction pathways. This may be a reasonable explanation for our observation that overexpression strains (*ptn1*^otef^ × *ptn1*^otef^) also displayed a reduction in virulence, and the haploid mutant cells were also more susceptible to cell wall stressors than the wild type. 

In *U. maydis*, infection of maize leads to the formation of galls or tumors that are filled with fungal hyphae that ultimately develop into teliospores [50]. Several individual processes inside the tumor lead to the formation of teliospores. These include the rounding and swelling of hyphal tips of fungal mycelia, for which functional *rum1* and *hda1* are necessary [60,61]. It has been shown that subsequent hyphal fragmentation would need *fuz1* [50]. Other studies have shown that several genes, including *cru1*, *hgl1*, *unh1*, *ust1*, and *ros1* influence the formation and maturation of teliospores, and some of the phenotypes of mutants in these genes resemble those shown by *ptn1* mutants. For instance, *cru1* mutants produced tumors that were considerably smaller and that contained very few teliospores [62]. On a similar note, *hgl1* mutants produced tumors, but failed to produce teliospores [63]. The deletion of *unh1* showed a defect in the formation, maturation, and pigmentation of teliospores [29]. A recent study indicated that the transcription factor Ros1 is a master regulator of sporogenesis in *U. maydis.* Although *ros1* is not required for tumor formation, it is essential during late stages of infection for fungal karyogamy, the proliferation of fungal hyphae, and spore formation [64]. The data from most of these studies indicate that the process of tumor formation is separate from teliospore development, as is also evident in the case of *ptn1* mutants. This makes *ptn1* part of the growing number of genes that are vital to the complex process of teliospore formation and germination.

In *Arabidopsis thaliana*, *AtPTEN1* (*PTEN* orthologue) is expressed exclusively in pollen grains during the late stage of development; when expression of AtPTEN1 was suppressed by RNA interference, it resulted in pollen cell death after mitosis [47]. This is interesting, considering the observation that the *ptn1* gene in *U. maydis* is involved in the sexual development of the fungus. Similarly, in *S. cerevisiae*, *TEP1* (*PTEN* orthologue) mutants showed a defect in the developmental process of sporulation. Moreover, *FgTep1p* (*PTEN* orthologue) deletion mutants in the wheat pathogen, *Fusarium graminearum*, produced fewer conidia and showed an attenuation in virulence on host plants [65].

In conclusion, the results reported in this study illustrate that *ptn1* has a role in virulence, especially in tumor production and in the development of teliospores in *U. maydis*. A growing number of studies show that the process of teliospore formation and germination is intricate, requiring exquisite control of hyphal proliferation inside the plant, the formation of teliospores, the formation of a thick cell wall, and germination involving the completion of meiosis, to produce haploid cells. The cells must make appropriate developmental decisions at each of these steps in response to the prevailing conditions. A defect in the genes responsible for any of these steps can result in failure to produce mature and functional teliospores. Hence, further investigation into the proteins that interact with Ptn1 during spore formation will likely contribute to a deeper understanding of how the complex signaling networks regulate pathogenic development and teliospore development in *U. maydis*. 

## Figures and Tables

**Figure 1 jof-05-00001-f001:**
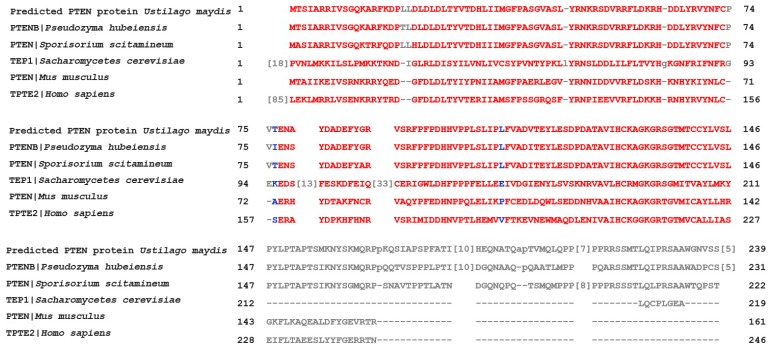
Sequence alignment of *U. maydis* Ptn1 and other PTEN proteins. Ptn1 protein was compared to selected proteins from Basidiomycete fungi *Pseudozyma hubeiensis* and *Sporisorium scitamineum*; Ascomycete fungi, *Saccharomycetes cerevisiae*; and the mammals, *Mus musculus* and *Homo sapiens*. The red color shows a high degree of sequence similarity in amino acids at the N-terminal regions between Ptn1 and the other identified proteins.

**Figure 2 jof-05-00001-f002:**
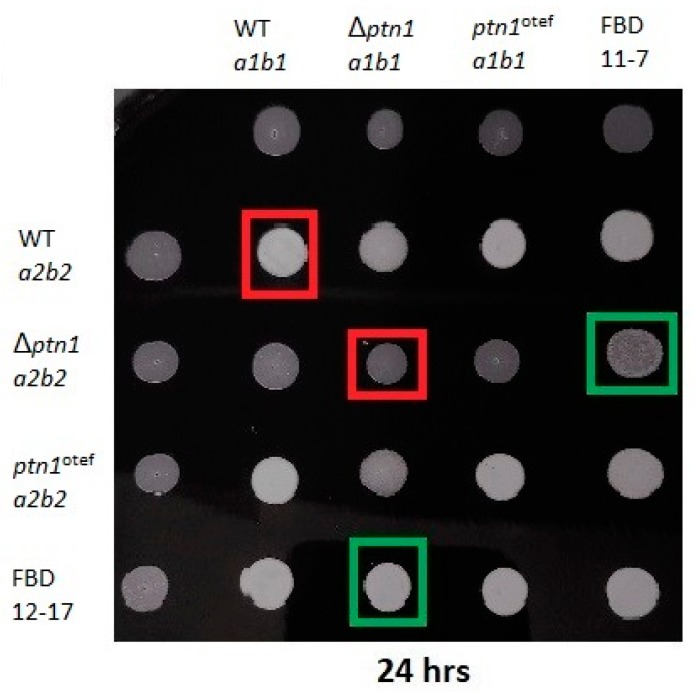
Mating assay and testing for pheromone production. Charcoal plate mating assay with wild type, Δ*ptn1*, Δ*ptn1*c, and FBD-tester strains. A positive mating reaction is indicated by a white “fuzz” phenotype of aerial hyphae production. The reduction in filamentation was noticeable at 24 h, while such differences disappeared by 48 h (not shown). Red boxes compare the ‘fuzz’ produced between wild type mating and Δ*ptn1* mating. Green boxes indicate the ‘fuzz’ produced when Δ*ptn1* strains were mixed with tester strains.

**Figure 3 jof-05-00001-f003:**
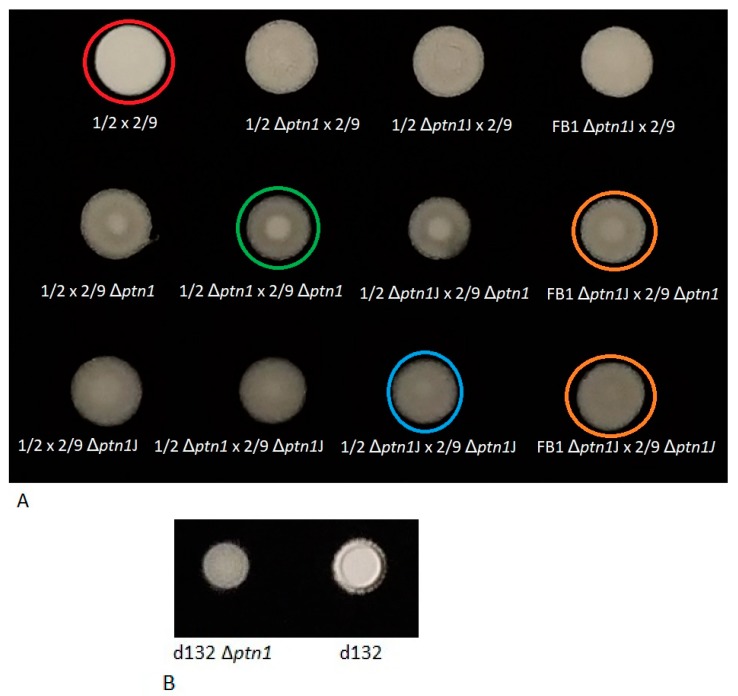
(**A**) Mating compared between deletion strains using two different constructs and different genetic backgrounds. Compared to wild type mating (1/2 × 2/9) denoted by the red circles, mating of deletion strains (1/2 Δ*ptn1* × 2/9 Δ*ptn1*, green circle, as well as 1/2 Δ*ptn1*J × 2/9 Δ*ptn1*J, blue circle) showed reduction in mating at 24 h. This result matched the observation made when a haploid deletion strain Δ*ptn1*J in the FB1 background was mated with a haploid Δ*ptn1* or Δ*ptn1*J strain in the 2/9 background (indicated by orange circles). (**B**) Formation of aerial hyphae for wild type *ptn1ptn1* or mutant *ptn1*Δ*ptn1* diploid d132 background at 24 h.

**Figure 4 jof-05-00001-f004:**
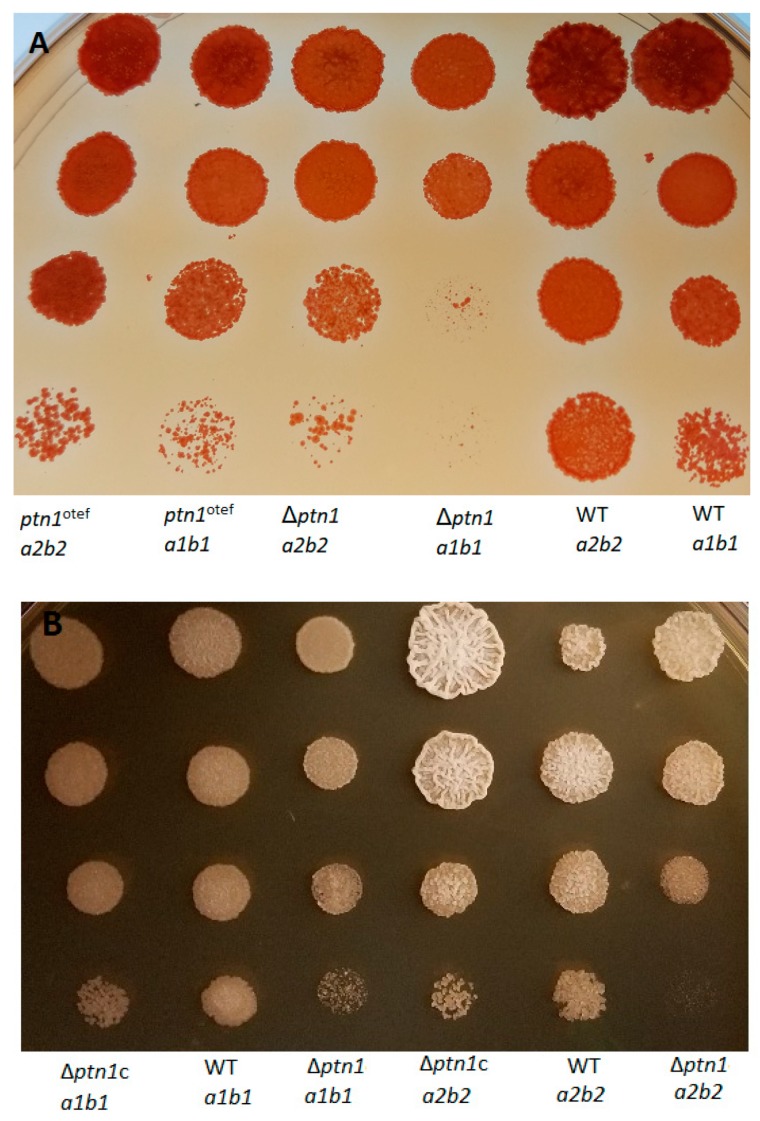
Response of mutants to cell wall stress. (**A**) 10-fold serial dilutions of the 1/2 *a1b1*, 2/9 *a2b2*, Δ*ptn1 a1b1*, Δ*ptn1 a2b2*, *ptn1*^otef^
*a1b1*, and *ptn1*^otef^
*a2b2* strains were spotted onto media containing 1 mM Congo Red, and observations were made at 48 h. Congo Red is a cell wall stressor which prevents glycan microfibril assembly by binding to β-glucans. The 1/2 *a1b1* strains were compared with *a1b1* mutant strains, while *a2b2* background strains were compared amongst themselves. Growth defects were observed in the Δ*ptn1* strains, as well as *ptn1*^otef^, as reductions in colony numbers at higher dilutions compared to comparable wild type, for both the *a1b1* and *a2b2* backgrounds. The overexpression of *ptn1* ectopically in deletion mutants was able to partially rescue the growth defect in (**B**) Congo Red media and (**C**) calcofluor white media.

**Figure 5 jof-05-00001-f005:**
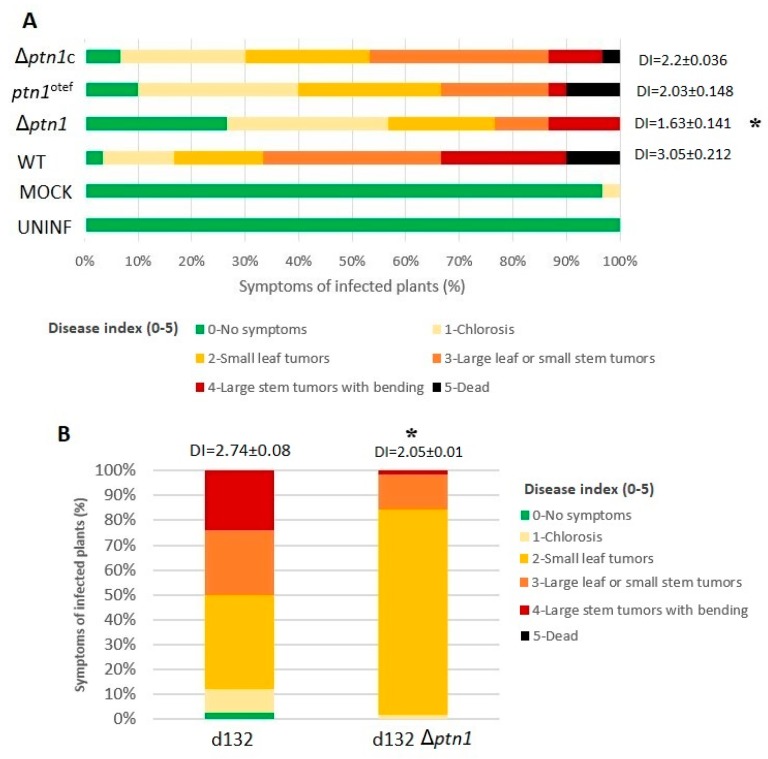
Plant infection. (**A**) One-week-old maize seedlings were infected with indicated strains of *U. maydis*. Disease symptoms were scored 14 days post-infection (dpi), and compared to those where both partners were wild type (WT) strains (1/2 *a1b1* × 2/9 *a2b2*). Plants were inoculated with one paired background, indicated on the Y-axis. Disease symptom categories are depicted at the bottom. Numbers to the right of each bar represent mean (±standard deviation) for disease indices (DI). The graphs display the percentage of plants with specific symptoms of infection. The Δ*ptn1 (a1b1* × *a2b2)* and *ptn1*^otef^ (*a1b1* × *a2b2*) infections; here, both partners contained the indicated mutations/modifications, displayed reduced pathogenicity. The reduction in Δ*ptn1* strains was statistically significant (*p* < 0.05) compared to the wild type used in this study. (**B**) Experiments conducted using diploid strains d132 WT and d132 Δ*ptn1* showed that the deletion of one copy of *ptn1* reduces pathogenicity. Numbers above each bar represent mean (± standard deviation) for disease indices (DI). The data were analyzed using Kruskal–Wallis test, followed by a post hoc comparison, and an asterisk (*) indicates significant difference (*p* < 0.05).

**Figure 6 jof-05-00001-f006:**
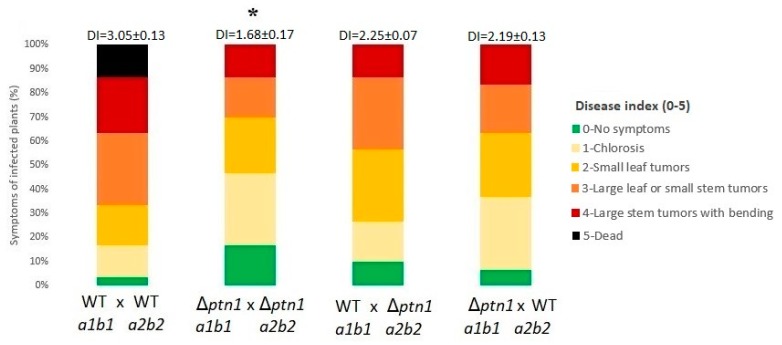
Are functional copies of the *ptn1* gene in both partners required for pathogenicity? The Δ*ptn1* strains were mated with the wild type strain of the opposite mating-type and compared with infections due to either both wild type or both mutant pairings. The graphs display the percentage of plants with specific symptoms of infection. The combinations of 1/2 Δ*ptn1 a1b1* × 2/9 *a2b2* and 1/2 *a1b1* × 2/9 Δ*ptn1 a2b2* were more virulent when compared to Δ*ptn1* (*a1b1* × *a2b2*), but less that when compared to 1/2 × 2/9. The numbers above each bar represent the mean (±standard deviation) for disease indices (DI). The data were analyzed using the Kruskal–Wallis test, followed by a post hoc comparison, and an asterisk (*) indicates significant difference (*p* < 0.05).

**Figure 7 jof-05-00001-f007:**
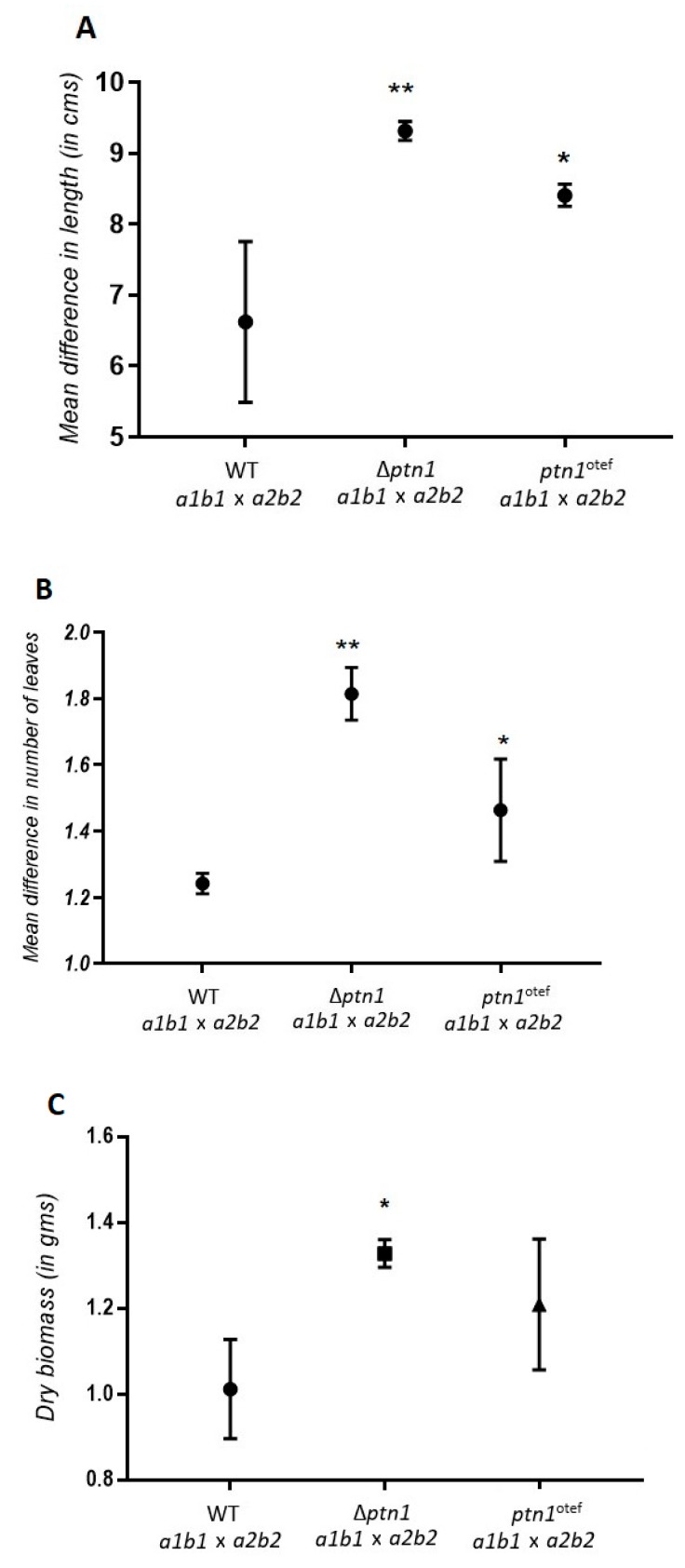
Plant growth parameters. Effect on (**A**) plant length, (**B**) leaf count, and (**C**) plant dry weight. The plants were infected with 1/2 *a1b1* × 2/9 *a2b2,* Δ*ptn1* (*a1b1* × *a2b2*) and *ptn1*^otef^ (*a1b1* × *a2b2*) and the length and the number of leaves produced were measured on dpi-0 (0 days post-infection), dpi-4 and dpi-8. The Δ*ptn1* strains showed a significant reduction in their effect on plant length and leaf production when compared to WT. Means (± standard deviation) of plant lengths and the number of leaves of Δ*ptn1* marked by an asterisk (*) were significantly different (*p* ≤ 0.05) or (**) were significantly different (*p* ≤ 0.01) from those of WT; the mean length and number of leaves of *ptn1*^otef^ was significantly different (*p* ≤ 0.05) from WT and Δ*ptn1*, respectively. The dry biomass (± standard deviation) of the plants infected with WT (*a1b1* × *a2b2),* Δ*ptn1 (a1b1* × *a2b2*) and *ptn1*^otef^
*(a1b1* × *a2b2*) were measured. The Δ*ptn1* strains showed an increase in plant biomass when compared to WT*,* indicating a reduction in virulence. Mean of dry mass of plants infected by Δ*ptn1* was significantly different (*p* ≤ 0.05) from those of wild type infections. Statistical analysis was done by ANOVA followed by Tukey’s test.

**Figure 8 jof-05-00001-f008:**
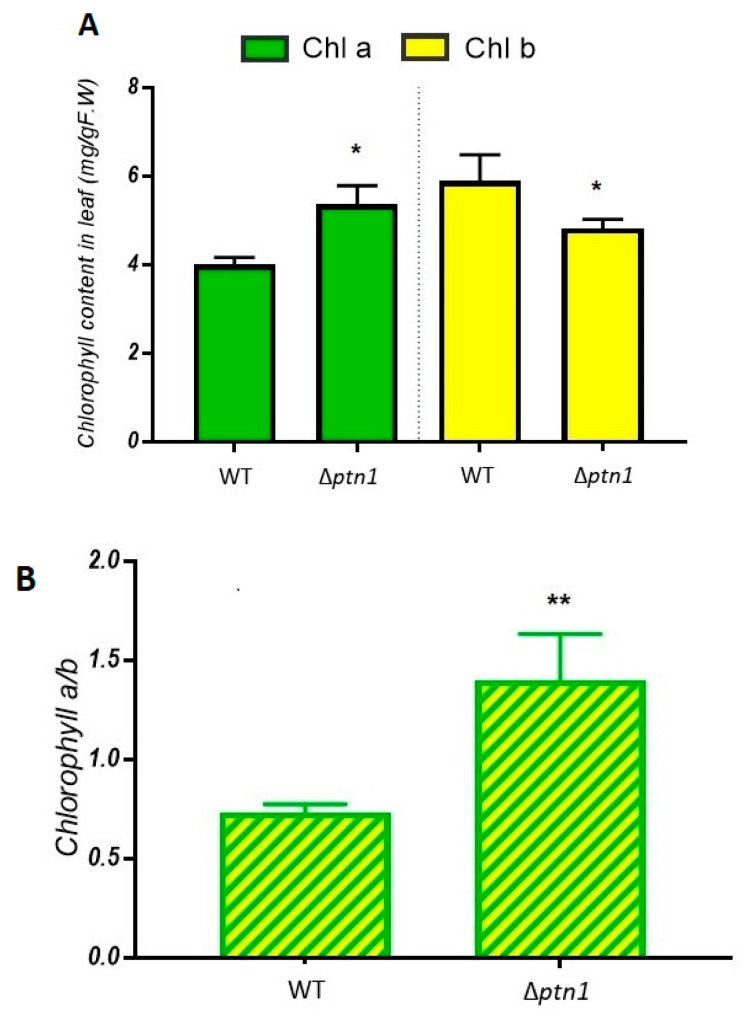
Chlorophyll estimation. (**A**) Chlorophyll count of leaves of plants infected with WT (1/2 *a1b1* × 2/9 *a2b2*), Δ*ptn1* (*a1b1* × *a2b2*) and *ptn1*^otef^ (*a1b1* × *a2b2*) were measured using the DMSO method [31]. (**B**) The plants infected with the Δ*ptn1* strains had higher *a*/*b* ratios, and such plants had chlorophyll levels that were more comparable to “mock” infections. Means of chlorophyll *a*, chlorophyll *b*, and chlorophyll *a*/*b* ratio were marked with an asterisk (*) or two asterisks (**) were significantly different (*p* ≤ 0.05) and (*p* ≤ 0.01), respectively, by Tukey’s test.

**Figure 9 jof-05-00001-f009:**
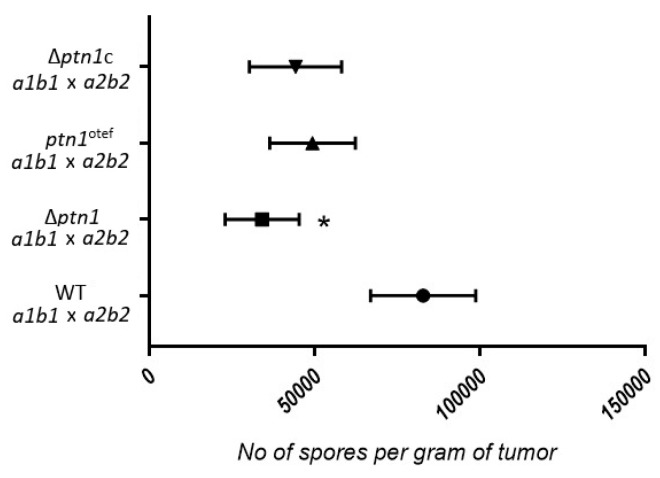
Tumors were collected from plants infected with 1/2 *a1b1* × 2/9 *a2b2*, Δ*ptn1* (*a1b1* × *a2b2*)*, ptn1*^otef^ (*a1b1* × *a2b2*) and Δ*ptn1*c (*a1b1* × *a2b2*). From each type of infection (e.g., 1/2 *a1b1* × 2/9 *a2b2*) tumors were pooled and used to extract spores. The spore count was obtained using a hemocytometer for each pooled extraction. Presented here are the mean spore count (± standard deviation) for three biological replicates of each infection experiment. There was a significant reduction in spore production in infections by the Δ*ptn1* strains, indicated by * (*p* < 0.05, by Tukey’s Test).

**Figure 10 jof-05-00001-f010:**
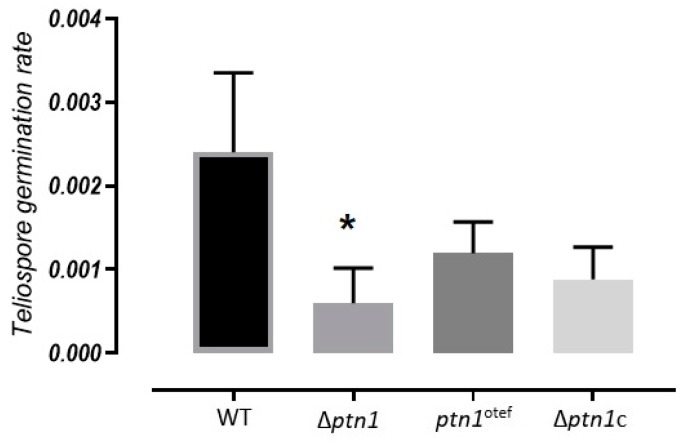
Spore germination assay. Spores were extracted from the plant tumors produced from the infection of 1/2 *a1b1* × 2/9 *a2b2,* Δ*ptn1 a1b1* × Δ*ptn1 a2b2, ptn1*^otef^
*a1b1* × *ptn1*^otef^
*a2b2*, and Δ*ptn1*c *a1b1* × Δ*ptn1*c *a2b2*, and plated onto water agar plates. The plates were incubated at 30 °C for 72 h, and observed for the presence of colonies formed from the germination of spores. Spore germination was reduced in Δ*ptn1 a1b1* × Δ*ptn1 a2b2*, and *ptn1*^otef^
*a1b1* × *ptn1*^otef^
*a2b2* combinations, with few colonies being observed after germination. The difference in germination rate (percentage germination per spores plated (± standard deviation) for Δ*ptn1* was statistically significant, indicated by * (*p <* 0.05) by Tukey’s Test; the differences for the overexpression strain were not statistically significant.

**Figure 11 jof-05-00001-f011:**
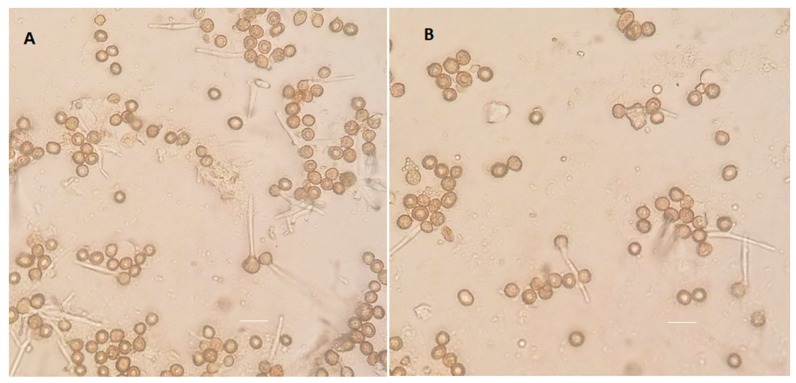
Spore germination tube formation. Microscopic analysis of spores collected from tumors of plants infected with (**A**) 1/2 *a1b1* × 2/9 *a2b2* and (**B**) Δ*ptn1 a1b1* × Δ*ptn1 a2b2* strains to examine possible defects in the formation of germination tube. Spore germination on water agar plates incubated at 29 °C for 30 h was examined using differential interference contrast (DIC) microscopy. Scale bars, 20 μm.

**Table 1 jof-05-00001-t001:** List of strains.

*Ustilago* Strain	Genotype ^a^	References
1/2	*a1b1*	[19]
2/9	*a2b2*	[19]
1/2 Δ*ptn1*	*a1b1 ptn1::cbx^R^*	This study
1/2 Δ*ptn1*J	*a1b1 ptn1::hyg^R^*	This study
2/9 Δ*ptn1*	*a2b2 ptn1::cbx^R^*	This study
2/9 Δ*ptn1*J	*a2b2 ptn1::hyg^R^*	This study
FB1 Δ*ptn1J*	*a1b1 ptn1::hyg^R^*	This study
1/2 *ptn1*^otef^	*a1b1 P_otef_-ump2, cbx^R^*	This study
2/9 *ptn1*^otef^	*a2b2 P_otef_-ump2, cbx^R^*	This study
SG200	*a1mfa2 bE1bW2*	[20]
SG200 Δ*ptn1*	*a1mfa2 bE1bW2 ptn1::hyg^R^*	This study
d132	*a1a2b1b2*	[21]
d132 Δ*ptn1*	*a1a2b1b2 ptn1:: hyg^R^*	This study
1/2 Δ*ptn1*c	*a1b1 ptn1::cbx^R^ P_otef_-ptn1, hyg^R^*	This study
2/9 Δ*ptn1*c	*a2b2 ptn1::cbx^R^ P_otef_-ptn1, hyg^R^*	This study
FBD11-7	*a1a1 b1b2*	[16]
FBD12-17	*a2a2 b1b2*	[16]
**Plasmids**	**Genotype**	**References**
DelsGate *ptn1*	*kan^R^ ptn1::cbx^R^*	[22]
Otef-*ptn1-cbx*	*bla(amp^R^) P_otef_-ptn1 cbx^R^*	This study
Otef-*ptn1-hyg*	*bla(amp^R^) P_otef_-ptn1 hyg^R^*	This study

^a^*cbx^R^*, carboxin resistance; *hyg^R^*, hygromycin resistance; the otef promoter is a modified tef promoter in which two direct repeats of a synthetic fragment containing seven tetracycline-responsive elements precede the promoter.

**Table 2 jof-05-00001-t002:** List of primers.

Primers	Sequence (5’-3’)
UPTN-K1	TAGGGATAACAGGGTAATCAACAAAATGACGCAACCAC
UPTN-K3	GGGACAAGTTTGTACAAAAAAGCAGGCTAAGAACGATGTTTGCGTGTCAG
UPTN-K4	GGGGACCACTTTGTACAAGAAAGCTGGGTACGTTAGACAAAGCGCAATCA
UPTN-K2	GGAAGGTGATCGTGAAGGAA
PTENUPLT	GCAACAAAATGACGCAACC
PTENUPRT	GTAGTTACCACGTTCGGCCATAAGGTGGTGGGACTGCTTT
PTENDNLT	GCTTTGCTCTTTGCTCTTCTG
CPDNRT	GCCTTAATTAATATACGCCCGATTGCTTACC
HYGPTENLT	AAAGCAGTCCCACCACCTTATGGCCGAACGTGGTAACTAC
HYGPTENRT	AGAAGAGCAAAGAGCAAAGCCTCAGGCCTCATGTTTGACA
PTEN5	TGAGGCCTGAGTGGCCTATATACCCCCTGCCCCTGT
PTEN3	CGAATTCTTGCGCTTTGTCTAACGAAC
HYGSPH1RTNEW	TTTGCATGCCTCAGGCCTCATGTTTGACA
HYGMFE1	TTTCAATTGTGGCCGAACGTGGTAACTAC

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
