# Peer review of "Deletion of ptn1, a PTEN/TEP1 Orthologue, in Ustilago maydis Reduces Pathogenicity and Teliospore Development"

_jof, 2018, doi:10.3390/jof5010001_

Round 1
Reviewer 1 Report
In this manuscript, the authors generated the deletion and overexpression strains of ptn1 gene in corn smut fungus Ustilago maydis and characterized the phenotypes. ptn1 is a putative ortholog of PTEN, which encodes phosphatase to dephosphorylate phosphoinositide trisphosphate and is known as tumor suppressor gene in mammals. Although overexpression of ptn1 does not provide detectable phenotypes on fungus, the deletion of ptn1 results in the reduction of mating efficiency, virulence and teliospore formation. Thus, the authors claim that PTEN ortholog in U. maydis is a virulence factor.
Since it has been already reported that functional ortholog of PTEN plays a role in virulence of plant pathogenic fungus Fusarium graminearum as the authors cited in the manuscript, I would consider that scientific novelty of the manuscript is not high. However, the manuscript is clearly written and the data presented supports the conclusions of authors. The deletion mutant of ptn1 in U. maydis shows similar phenotypes to the precedent study in F. graminearum, which support that the results given by the authors are reliable. Therefore, I would like to point out only minor corrections in text.
In entire text, the authors should take consistency for words (e.g. wildtype or wild type)
Page 2, Line 49; suggests -> suggest
Page 3, Line 88; Potef--based… -> Potef-based
Page 4, Line118; OD600 of 3 -> OD600 of 3.0
Author Response
Since it has been already reported that functional ortholog of PTEN plays a role in virulence of plant pathogenic fungus Fusarium graminearum as the authors cited in the manuscript, I would consider that scientific novelty of the manuscript is not high.
We respectfully disagree with the Reviewer as to the scientific novelty of the finding that a PTEN ortholog plays a virulence role in U. maydis. While both the orthologues of F. graminearum and U. maydis play roles in virulence, the former is involved in asexual spore (conidia) production, but there is no effect on germination frequency, unless wortmannin, the phosphatidylinositol-3 kinase inhibitor, is present (Zhang et al. Molecular Plant Pathology 11(4):495-502). In contrast, deletion of U. maydis ptn1 reduces several aspects of virulence for this fungus, including tumor production, sexual spore (teliospore) development, and rate of teliospore germination. These important roles for Ptn1, places this ortholog within a growing group of proteins in U. maydis that playing varying parts in spore development.
However, the manuscript is clearly written and the data presented supports the conclusions of authors. We thank the Reviewer for this assessment.
The deletion mutant of ptn1 in U. maydis shows similar phenotypes to the precedent study in F. graminearum, which support that the results given by the authors are reliable. Therefore, I would like to point out only minor corrections in text.
In entire text, the authors should take consistency for words (e.g. wildtype or wild type) We have made this change.
Page 2, Line 49; suggests -> suggest We have made this change.
Page 3, Line 88; Potef--based… -> Potef-based We have made this change.
Page 4, Line118; OD600 of 3 -> OD600 of 3.0 We have made this change.
Reviewer 2 Report
The manuscript “Deletion of ptn1, a PTEN/TEP1 orthologue, in Ustilago maydis reduces pathogenicity and teliospore development” by Vijayakrishnapillai et al., describes the deletion of ptn1 a component/regulator of the mTOR pathway and its subsequent analysis of phenotypes mainly regarding mating, virulence and sporulation.
The manuscript is in general well written and it is mostly clear in the presentation of the data. The findings are interesting and help to understand the contribution of this important pathway to virulence and sporogenesis in U. maydis.
However, here are some concerns which should be addressed prior to publication:
- What is the expression of ptn1 in the overexpression and the complemented strains? Is it similar to wt or higher or lower? This could explain some of the phenotypes. Also for the complementation the none native promoter was used. Actually the same as for the overexpression strain only at a different location in the genome. Thus the complemented strains also could be a overexpression strain.
- How were the deletion strains confirmed? I didn’t see any reference to southern hybridisation in MM or results.
- are the data with standard deviation or standard error?
- Figure 3: as FB1 is a a2b2 strain (according to authors, Table1) how can it mate with 2/9 which also is a a2b2 strain?
- qPCR: I didn’t see any statistics for this analysis? (Fig. S5)
- Were any other stressors tested, for example H2O2, azole drugs,…? Or different carbon sources? As the mTOR pathway contributes to resistance to stressors and usage of other carbon sources (as mentioned by the authors). This should be tested.
In case there is a connection with stressors, there might be a connection of ptn1 with mitochondrial functions, apoptosis and mating (Kretschmer et al., 2018 “Acetate provokes mitochondrial stress and cell death in Ustilago maydis”).
- Fig5: the text needs to be checked and clarified. (Plants were inoculated with one paired background, indicated on the X-axis?; line 281infections, here both partners?
- Figure 7 shows curves for A and B and only data points for C. In my opinion only data points or maybe bars should be used, as there is no continuum of data between the 3 samples (for example as seen for time points)
For Figure 7 the data are from 2 biological replicates according to section 3.6. How was a statistical analysis possible? As this analysis is not very common for U. maydis I would recommend to use caution for the interpretation of the data.
- U. maydis wt infection is known to reduce chlorophyll content and also other pigments. In Figure 8 no uninfected control is shown, it would be nice to cite a paper which already published the effect of chlorophyll content of control vs. infection (There are several option such as Kretschmer et al., or Doehlemann et al….)
- Section 2.4 mentions cell viability. How was cell viability measured and what were the results? Was a tunel assay used to estimate apoptosis or cell viability staining with a dye?
Minor:
-line 65 um03760 is the old genome designation maybe the new one should be named somewhere in the manuscript
-line 78 what was the concentration of CFW
-section 2.4: is YPS the same as YEPS?
In line 110: 10^8 cells were used in 10ml leading to 10^7cells per ml. This is very high and should equal an OD of by far greater than 1. Ustilago in PDB growths to around 10^8 cells per ml as maximum. Is this section correct? It doesn’t seem to fit with Figure S6.
If for the infection a DI was calculated (as mentioned here) it should be shown in the figures such as Fig 5 and 6
-Fig.1 the red colour should be a brighter red as it showed more pink on my PC
-line 194 was compared with the (delete) that
-line 221. actually there are also compounds like acetate which inhibit mating and the formation of fuzz
- The authors use very often the term to make or made, there might be better ways for example line 268 to form tumors, line 82 strains were made maybe better created,…
-Fig.10: the germination rates is expressed as percent? It appears pretty low if it is the case.
-line 425 in imparting pathogenicity = Impairing? This sentence needs clarification
Author Response
- What is the expression of ptn1 in the overexpression and the complemented strains? Is it similar to wt or higher or lower? This could explain some of the phenotypes. Also for the complementation the none native promoter was used. Actually the same as for the overexpression strain only at a different location in the genome. Thus the complemented strains also could be a overexpression strain.
– We did not examine ptn1 expression for overexpression strains or the complemented strains. In part this was due to the lack of prominent phenotypes compared to wild type for the overexpression strain. We did observe subtle defects during infection and on spore germination. We discussed these findings citing references where overexpression may sometimes yield a similar phenotype as deletion. The Reviewer is correct that the complemented strains are under the control of otef promoter. However, the original overexpression strains also have the original copy of the ptn1 gene under control of its native promoter. Hence a possible reason for the lack of complete rescue of the phenotype could be that the native promoter is important in fully regulating its function.
- How were the deletion strains confirmed? I didn’t see any reference to southern hybridization in MM or results. The deletion of the gene for all strains was confirmed using judiciously chosen primers for PCR and further confirmed by qRT PCR. In all the deletion mutants, the wild type amplicon was missing (while present in wild type) and combinations of primers chosen from within and outside the construct produced the expected sized bands in the putative deletion strains. Moreover, the deletion strains failed to show expression (i.e., amplified cDNA target) in qRT-PCR.
- are the data with standard deviation or standard error?
The data are shown with standard deviation. (Graphs showing biomass, plant length, leaves number, chlorophyll content, chlorophyll ratio, spore number, spore germination, q-RT – standard deviation)
- Figure 3: as FB1 is a a2b2 strain (according to authors, Table1) how can it mate with 2/9 which also is a a2b2 strain? This was mislabeled in Table 1. It has been changed to a1b1.
- qPCR: I didn’t see any statistics for this analysis? (Fig. S5) – The RNA expression data were analyzed in GraphPad Prism using unpaired T-test after normalization to wild type expression under high ammonium conditions and a probability value of P < 0.05 was considered as significant. This is now indicated in Materials and Methods and in the legend to Fig. S5.
- Were any other stressors tested, for example H2O2, azole drugs,…? Or different carbon sources? As the mTOR pathway contributes to resistance to stressors and usage of other carbon sources (as mentioned by the authors). This should be tested. We tested the effect of H2O2. The ptn1 deletion strains and wild-type strains were allowed to grow for 24-48 hours at 26 oC on YEPS agar plates supplemented with 0.8 mM H202 to examine the effect of H202 on the strains. However, we didn’t observe any difference between the wild type and Δptn1.
In case there is a connection with stressors, there might be a connection of ptn1 with mitochondrial functions, apoptosis and mating (Kretschmer et al., 2018 “Acetate provokes mitochondrial stress and cell death in Ustilago maydis”). Thank you suggesting this idea. However, we have not yet tested acetate or other carbon sources as stressors. In future, we think that would be a useful set of experiments.
- Fig5: the text needs to be checked and clarified. (Plants were inoculated with one paired background, indicated on the X-axis?;
For Fig. 5A the Reviewer is correct, the axes were indicated incorrectly; we should have said Y- axis. This has now been corrected.
line 281infections, here both partners? We have reworded the sentence to indicate more clearly that both partners bore the deletion or both were complemented strains.
- Figure 7 shows curves for A and B and only data points for C. In my opinion only data points or maybe bars should be used, as there is no continuum of data between the 3 samples (for example as seen for time points) We changed the graph showing only data points.
For Figure 7 the data are from 2 biological replicates according to section 3.6. How was a statistical analysis possible? The original wording was that the infection experiments were carried out and then repeated twice (i.e., 2 more times). We have changed the wording to now indicate that the infections were each performed a total of three times.
As this analysis is not very common for U. maydis I would recommend to use caution for the interpretation of the data. Even though this analysis (leaf number, plant height, biomass as measures of virulence) is not common for U. maydis, it has been performed in examining pathogenicity of other fungal species (Epichloe: Zabalgogeazcoa et al. European Journal of Agronomy 24:374-384; Austropuccinia psidii:Winzer et al. 2017. Austral Ecology 43:56-68.) Also, this is just another way to look at the level of infection in the plant caused by the different fungal strains. We have measured the pathogenicity by the traditional method as well.
- U. maydis wt infection is known to reduce chlorophyll content and also other pigments. In Figure 8 no uninfected control is shown, it would be nice to cite a paper which already published the effect of chlorophyll content of control vs. infection (There are several option such as Kretschmer et al., or Doehlemann et al….) We now include a statement indicating this effect in wild type infections by U. maydis and cite the following references:
Doehlemann, G.; Wahl, R.; Horst, R.J.; Voll, L.M.; Usadel, B.; Poree, F.; Stitt, M.; Pons-Kühnemann, J.; Sonnewald, U.; Kahmann, R., et al. Reprogramming a maize plant: transcriptional and metabolic changes induced by the fungal biotroph Ustilago maydis. The Plant Journal 2008, 56, 181-195, doi:10.1111/j.1365-313X.2008.03590.x.
Horst, R.J.; Engelsdorf, T.; Sonnewald, U.; Voll, L.M. Infection of maize leaves with Ustilago maydis prevents establishment of C4 photosynthesis. Journal of plant physiology 2008, 165, 19-28, doi:10.1016/j.jplph.2007.05.008.
- Section 2.4 mentions cell viability. How was cell viability measured and what were the results? Was a tunel assay used to estimate apoptosis or cell viability staining with a dye? We did not measure cell viability. The heading and description are now changed. We just measured growth on agar plates or measure growth rates in broth.
Minor:
-line 65 um03760 is the old genome designation maybe the new one should be named somewhere in the manuscript. This has been updated to reflect the new number UMAG_03760 at two places.
-line 78 what was the concentration of CFW- The concentration of 50 µM calcofluor white is now stated in the Materials and Methods section
-section 2.4: is YPS the same as YEPS? This has been changed to YEPS throughout the article.
In line 110: 10^8 cells were used in 10ml leading to 10^7cells per ml. This is very high and should equal an OD of by far greater than 1. Ustilago in PDB growths to around 10^8 cells per ml as maximum. Is this section correct? It doesn’t seem to fit with Figure S6.
The amount used was misstated. It should read (and now has been corrected) “4 x 106 cells were added into 10 ml.”
If for the infection a DI was calculated (as mentioned here) it should be shown in the figures such as Fig 5 and 6. All the three graphs 5a,5b,and 6 have been updated to reflect disease index +/- standard deviation.
-Fig.1 the red colour should be a brighter red as it showed more pink on my PC- The picture is updated to reflect a dark red color
-line 194 was compared with the (delete) that. This has been corrected.
-line 221. actually there are also compounds like acetate which inhibit mating and the formation of fuzz. We now mention that acetate inhibits filamentation due to mating and due to exposure to oleic acid and have added the reference:
Kretschmer, M.; Lambie, S.; Croll, D.; Kronstad, J.W. Acetate provokes mitochondrial stress and cell death in Ustilago maydis. Mol Microbiol 2018, 107, 488-507, doi:10.1111/mmi.13894.
- The authors use very often the term to make or made, there might be better ways for example line 268 to form tumors, line 82 strains were made maybe better created
These have been modified as per the Reviewer’s suggestions.
-Fig.10: the germination rates is expressed as percent? Yes. We have recalculated the germination rates and updated the changes in the graph.
It appears pretty low if it is the case.
Under the conditions we have used, this is the germination we have seen.
-line 425 in imparting pathogenicity = Impairing? This sentence needs clarification. We have changed the sentence to indicate the role of ptn1 in the pathogenic program of U. maydis.
Round 2
Reviewer 2 Report
Dear Authors,
Thank you for the improved version of your manuscript.
In general it is fine now. I only have two suggestions/concerns: The missing expression of ptn1 in the mutants, as this could have given answers about the importance of the native promoter and the importance of the right number of transcript/protein for the proper function. But the explanation in the discussion is probably good enough.
Second, thanks for including the info about the mutant construction and confirmation. I agree that the PCR and qPCR results indicate the deletion of the gene. However, secondary ectopic integration of the construct can’t be excluded with those analyses. Southern hybridization or whole genome sequencing of the mutants would be acceptable.
The partial complementation and especially the deletion in several genetic backgrounds make a case for the observed phenotypes to be related to ptn1. Thus I would consider it ok as it is for this publication.